# Neurodevelopmental Outcomes among Brazilian Children with Cyanotic Congenital Heart Disease and Its Associated Factors

**DOI:** 10.3390/medicina58111669

**Published:** 2022-11-18

**Authors:** Flávia Saraçol Vignol, Priscila Aikawa, Tatiane Britto da Silveira, Ronan Adler Tavella, Vinita Mahtani-Chugani, Emílio J. Sanz, Flavio Manoel Rodrigues da Silva Júnior

**Affiliations:** 1Faculty of Medicine, Universidade Federal do Rio Grande—FURG, Rio Grande 96203-900, Brazil; 2School Hospital, Universidade Federal de Pelotas, Pelotas 96020-360, Brazil; 3Unidad de Investigación HUNSC y Gerencia de Atención Primaria de Tenerife, 38200 Santa Cruz de Tenerife, Spain; 4Department of Physical Medicine and Pharmacology, Universidad de La Laguna, 38200 Santa Cruz de Tenerife, Spain

**Keywords:** heart diseases, child development, education, cyanosis

## Abstract

*Objectives:* The aim of this study was to evaluate neurodevelopmental outcomes (motor development, nonverbal intelligence, and attention) in children with cyanotic congenital heart disease (CHD) compared with healthy children from a public hospital in southern Brazil. *Materials and Methods:* This was a cross-sectional study with pediatric patients of both sexes: 37 children with cyanotic CHD and a control group with 38 healthy children. Parents/guardians undertook a questionnaire and the SNAP IV scale (to evaluate attention) was applied. Two instruments were applied to each child: the R-2 Non-Verbal Intelligence test and the motor development scale. To assess the factors associated with insufficient performance in the three fields of neurodevelopment, a Poisson regression analysis was performed with a robust estimate. *Results:* There were no significant differences between children with cyanotic CHD and the control group for any of the neurodevelopmental outcomes studied. Low socioeconomic class was a factor associated with worse performance on the intelligence test and inattention. Furthermore, age was a factor for performance on the intelligence test, while a greater number of siblings was a factor associated with worse performance on the attention test. *Conclusions:* Public policies regarding child health must involve prioritizing the improvement of families’ social conditions.

## 1. Introduction

Cardiovascular structural and functional abnormalities present at birth are defined as congenital cardiac malformations. In most cases, they result from embryonic alterations of the normal primitive heart structure or from incomplete or insufficient development [1]. Congenital malformations are among the main causes of mortality in early childhood, with congenital heart diseases (CHDs) representing 40% of these malformations [2]. Most cardiac anomalies are of unknown etiology; however, some factors are associated with higher incidence, such as the mother being over 40 years old and other genetic factors [3].

CHDs are clinically classified as acyanotic and cyanotic, the latter being manifested by bluish skin due to insufficient blood oxygenation or changes in blood flow [4,5]. In Latin America, congenital heart defects are the second leading cause of death in children under the age of one, posing a significant public health problem [6,7].

The worldwide prevalence of CHD is approximately 8/1000 live births, and one-third will need surgery during the first year of life [8]. In Brazil, approximately 23,000 children are born with heart problems per year, and of these, 80% will need surgical treatment. However, it is estimated that approximately 13,000 children do not receive adequate treatment due to failure in diagnosis or a lack of access to the public health system [9].

In addition to mortality outcomes, neurodevelopmental impairment is common in children with moderate and severe CHD [10]. Several factors can significantly interfere with neuropsychomotor development [11], including biological, psychological, social, and environmental factors [12]. Adequate neurodevelopment is related to the sequence of processes that evolve over the chronological age of the individual, starting with simple and disorganized movements until the execution of highly organized and complex motor skills. It is characterized by motor, cognitive, and language learning obtained through the maturation of the nervous system and ordered through the experiences of the child [13].

Studies in developed countries were carried out with neonates and infants [14,15] and concluded delayed psychomotor development in CHD patients compared with a control group. In the Midwest and Northeast regions of Brazil, similar studies were carried out with neonates and infants, showing abnormal development in children with congenital heart disease, but these studies did not have a control group for a comparative evaluation [2,16]. The Brazilian context of CHD shows a situation of underreporting that can compromise the prognosis of patients, and in this sense, epidemiological studies should serve as a basis for health-planning actions [17].

Taking into account a set of studies previously carried out in other locations, our hypothesis is that the children with CHD included in this study had lower neuropsychomotor performance than healthy children with similar socioeconomic characteristics and residing in the same study region (control group). Thus, the aim of this study was to evaluate neurodevelopmental outcomes in different fields (motor development, nonverbal intelligence, and attention) in children with CHD at a public hospital in southern Brazil and analyze the factors associated with these outcomes compared with healthy children.

## 2. Materials and Methods

### 2.1. Research Design

A cross-sectional study was carried out with a group of children with CHD and a group of children without heart disease. The criteria for the diagnosis of CHD were based on clinical cardiological examinations and complementary tests, such as electrocardiograms (ECGs), chest X-rays, and pediatric echocardiography.

### 2.2. Sample and Study Site

The present study has the approval of the Ethics Committee in Research in the Health Area (CEPAS/Brasil) under number 3.746.217, following all of the ethical precepts of current Brazilian legislation. The free and informed consent form was accepted and signed by all parents/guardians. The children also signed the consent form. Anonymity and confidentiality were guaranteed for all participants, and all were informed that they could withdraw from the research at any time, without prejudice.

The study was conducted on a group of children with heart disease seen at the Pediatric Cardiology Outpatient Clinic and a group of children without heart disease seen at the Pediatric Outpatient Clinic. All children were patients at the Hospital Universitário Dr. Miguel Riet Correa Júnior, Rio Grande, southern Brazil. We recruited 75 children of both sexes aged between 5 and 11 years: 37 children diagnosed with CHD and 38 children without CHD.

The sample met the following inclusion criteria:(1)All study participants were between 5 and 11 years, 11 months old;(2)All participants were previously being followed up at the pediatric cardiology or pediatric outpatient clinics of the HU-FURG;(3)Patients with CHD were only included in the survey after confirmation of the diagnosis through pediatric Doppler echocardiography, (tetralogy of Fallot—37.8%; transposition of great vessels—18.9%; pulmonary atresia—16.2%; tricuspid atresia—13.5%; tricuspid atresia with RV hypoplasia—8.1%; and cyanotic complex heart disease—5.4%);(4)The volunteers in the control group came from the pediatric outpatient clinic, and only those who were exclusively in childcare consultations were included.

Exclusion criteria:(1)Participants who had associated syndromes that could interfere with the results of neurodevelopment tests or chronic cyanotic diseases were excluded;(2)Participants who were not within the age group established above were excluded;(3)Participants who did not complete all stages of the research were excluded.

### 2.3. Socioeconomic, Demographic, and Clinical Data

A semi-structured questionnaire was used with questions about demographic, socioeconomic, and clinical information adapted by the researchers from a pilot study. The questionnaire was administered by previously trained interviewers to the children’s parents or guardians. In this study, the variables included the age of the child (in months), socioeconomic class score according to the Brazilian Association of Research Companies (ABEP), sex of the child, number of siblings, reason for referral to the medical service, previous hospitalizations (yes/no), prenatal (yes/no), pulse oximetry (yes/no), and echocardiofetal (yes/no).

### 2.4. Dependent Variables

In this study, three dependent variables related to neurodevelopment were used: motor development, intelligence, and attention. These three instruments have been validated for the Brazilian population and are widely used in psychometric diagnosis.

#### 2.4.1. Motor Development

Motor development was assessed using the Motor Development Scale (MDS) developed by Rosa Neto [18]. This scale is used in Brazilian and international studies [19,20,21].

The children were individually referred to a room reserved for motor exercises. The exercises were performed with a maximum duration of 40 min for each child evaluated. Tests based on the child’s age were included to assess fine motor skills, global motor skills, balance, body scheme/rapidity, spatial organization, and laterality. The results of each were considered satisfactory or unsatisfactory, depending on whether the child correctly completed the test.

#### 2.4.2. Nonverbal Intelligence

The R-2 nonverbal intelligence test is based on the same theoretical principles as the Raven test and aims to measure the G-factor of intelligence (general factor) proposed by Spearman [22]. The model has been validated by the Brazilian psychology council [23] and is used both in Brazilian studies [24,25,26] and in international studies [27,28]. The test material consists of 30 boards with colored figures, identified as items, which are to be presented at one time to the child. The items are organized in increasing order of difficulty, consisting of geometric figures and objects common to the child’s experience. Each of them displays a figure with a part missing, and the child must identify the correct image among the alternatives available on the board. The figures are colored, with the aim of making the test more attractive and motivating for the child [23]. The R-2 Nonverbal Intelligence Test was applied to the participants individually in the respective pediatric cardiology and pediatric outpatient clinics under the supervision of a psychologist.

#### 2.4.3. Attention Deficit

The attention level assessment was performed using the first 9 items of the SNAP IV scale [29]. The SNAP IV scale is widely used [30,31,32] for screening attention deficit hyperactivity disorder in children; here, it was applied to parents and/or guardians in an appropriate room.

### 2.5. Data Analysis

The results were expressed as the mean ± standard deviation or were categorized as the frequency of occurrence. The tests applied (depending on the variables) were the t test (to compare the children’s average age) and the chi-square test to compare the frequencies of the other independent variables and dependent variables. To assess the factors associated with the three outcomes (motor development, intelligence, and attention), bivariate and multivariate Poisson regression analyses with robust estimation were used. The outcomes were dichotomized as follows: satisfactory/unsatisfactory motor development (when the child had an unsatisfactory performance in at least one test, it was considered unsatisfactory performance), intelligence deficits (children with lower-than-average performance according to the R2 test result were considered to have intelligence deficits), and inattention (children with at least 6 of the 9 items marked as “a lot” or “too much” were included in the inattention group). The critical *p* value for all analyses was 0.05, and analyses were also performed in SPSS 22 (Chicago, IL, USA).

## 3. Results

The results were organized considering the four instruments used in data collection: sociodemographic questionnaire, motor development scale, R-2 nonverbal intelligence test, and attention deficit test, as detailed below.

### 3.1. Sociodemographic Questionnaire

A total of 75 children were included in the study: 37 children with CHD and 38 children without CHD, of which 44 were boys (59%) and 31 were girls (41%), with no significant difference between the sexes in the total sample and no significant difference between groups (*p* = 0.39). Only one child who was unable to complete the tests was excluded, but his sociodemographic data were included. The mean age was 95.7 months for the CHD group and 99.3 months for the control group. Regarding the economic class, the highest percentage of participants were in the C1-C2 class, 70% in the CHD group and 87% in the control group, and there was a significant difference between classes (A-B1-B2 and D-E) (*p* = 0.01). Regarding the reason for the referral, 59% of patients with heart disease were referred for already established CC. We observed that 8 (22%) of the children with heart disease went to the Pediatric Cardiology Outpatient Clinic for other reasons, such as requesting a medical certificate for physical activities, and that 36 children (95%) without heart disease went to the Pediatric Outpatient Clinic for routine consultations. Not all pregnant women underwent a fetal echocardiogram; 45 pregnant women (78%) did not undergo a fetal echocardiogram while 13 (22%) underwent an exam. Among these 13 pregnant women who underwent fetal echocardiography, 9 babies were born with CC and 4 babies were born without CC (*p* = 0.06) Table 1).

### 3.2. Motor Development Scale

In the motor development scale (Figure 1), there was no significant difference between the groups (*p* = 0.77), and both groups presented a higher percentage of children with unsatisfactory motor tests. A total of 92.1% of children without heart disease had unsatisfactory performance in at least one test, while 83.7% of children with heart disease had unsatisfactory performance in at least one test.

The motor development scale had activities that measured ability in tests of fine motor skills, global motor skills, balance, body scheme/speed, spatial organization, and laterality, performed according to each age group, and children could make more than one attempt. The data with the frequencies of satisfactory results for each test of the motor development scale are shown in Table 2. Both groups of children performed worse in the speed test, but there was no significant difference in the correct frequency of the tests between children with and without heart disease.

In the motor development performance, there was no factor associated with this outcome among the factors included in the study (Table 3).

### 3.3. R-2 Nonverbal Intelligence Test

The results of the R-2 nonverbal intelligence test (Figure 2) showed a similar frequency between the two groups (*p* = 0.33), with a frequency of 74% of children in the control group being in the average or above average intervals, while this frequency among the CHD group was 62%.

Figure 3 shows the classification of non-verbal intelligence R-2 according to the degree of intelligence in all test classes: MH: much higher; U: upper; UM: upper mean; M: mean; LM: lower mean; B: borderline; and ID: intellectually disabled. There was no significant difference between the two groups of children (CHD and control) using this classification.

Crude and adjusted analyses of factors associated with worse performance in the non-verbal intelligence test are shown in Table 4, and the significant parameters in both analyses were age and socioeconomic class. Both age and better socioeconomic status were factors that reduced the prevalence ratio of worse performance in the non-verbal intelligence test.

### 3.4. Attention Deficit Test

The results for inattention are shown in Figure 4. The comparison between groups of children showed no significant differences between them, and the frequency of children classified with inattention was less than 20% (*p* = 0.96) (18.9%) among the children with CHD and 18.4% among children in the control group.

For the assessment of inattention (Table 5), better socioeconomic status was associated with a lower prevalence of inattention, while a greater number of siblings was associated with a higher prevalence of inattention. These findings were similar in the crude and adjusted analyses.

## 4. Discussion

This study aimed to evaluate possible differences between children with similar socioeconomic characteristics residing in the city of Rio Grande, Brazil, diagnosed with cyanotic CHD and without CHD (control group) in relation to aspects of motor development, non-verbal intelligence, and inattention. For the three main neurodevelopmental outcomes used in the study, there were no differences between the groups. Regarding the factors associated with higher prevalence of unfavorable outcomes, age, socioeconomic class, and the number of siblings were the significantly important variables.

Although our study did not find neurodevelopmental differences between children with and without cyanotic CHD, numerous studies have been conducted around the world that indicate worse neuro-psychomotor performance in children with heart disease. A group of German researchers conducted a study on children with CHD in need of cardiac surgery. The children were followed from birth to adulthood, and neurodevelopmental and cognitive deficits were observed in children of different age groups. In early childhood, motor and linguistic articulation deficit was observed; at school age, speech, attention, and memory were affected; in adolescence, executive, psychosocial, and psychiatric disorders, and impaired quality of life were observed; and in adulthood, neuro-cognitive, psychosocial problems, and professional perspectives were reported. In view of this, the German Society of Pediatric Cardiology started to require detailed serial neuropsychological examinations at 2 and 5 years of age, before puberty, and before adulthood, for the high-risk group of children with CHD operated on in childhood to detect and treat early stage partial-performance disorders [33].

In another study, Spanish researchers conducted an investigation with 54 children with severe CHD using the Denver test to investigate developmental delays at 2, 6, 12, 15, and 18 months of age, and concluded that the test performance was normal in 98.4% of patients at 2 months, 87.5% at 12 months, and 85% at 18 months. The authors reported a greater developmental delay among patients with more severe CHD who required cardiopulmonary bypass during the surgical procedure [14].

These results obtained from studies conducted in developed countries with better socioeconomic conditions were not reproduced in the present study. A reasonable line of reasoning to explain the findings of the present study is the social vulnerability of our study population that can impair or delay the development of psychomotor skills. In the Brazilian context, approximately 40% to 46% of children who are considered healthy present development evaluated as at risk and/or with suspected delay [16,34,35]. In a Brazilian study in the Northeast region that evaluated neuropsychomotor development in preschoolers without chronic diseases, a high prevalence of abnormal developmental performance was found, where 46.3% of the children studied did not adequately complete the tests [36].

The conditions at the beginning of life can be decisive for the evolution of the health–disease process and help to understand the inequalities between human groups with regard to illness throughout life [37]. Social vulnerability is closely linked to the unfavorable situations of certain population groups, characterizing an index of inequality in living conditions [38]. Socioeconomic, demographic, and environmental conditions are recognized factors associated with unfavorable health outcomes, and Brazilian studies have pointed out this relationship. In Brazil, the current federal constitution incorporated the social determinants of health (SDH) when it recognized that health is affected by housing, basic sanitation, the environment, work, income, education, leisure, and access to essential goods and services [39].

The studies mentioned above indicate that the family environment is related to the development of skills in healthy children, mainly variables related to the socioeconomic level, parents’ education, number of adults who live with the child, parents’ health, language stimulation, and interaction between parents and children. Our study was conducted with children from public schools who used the Brazilian public health service, and this may play a key role in the lack of difference between the two groups studied with regard to performance in the applied neurodevelopmental tests.

It is estimated that more than 200 million children under 5 years of age in low- and middle-income countries will not reach their developmental potential, mainly because of poverty, nutritional deficiencies, and inadequate learning opportunities [40]. Adequate neuropsychomotor development depends on intrinsic and extrinsic factors related to the child. The findings indicate that children at high risk and greater vulnerability are more likely to have a delay in neuropsychomotor development [41].

In Brazil, studies in children with CHD and assessments of psychomotor development are cross-sectional, observational studies, without a control group, on children aged 1 month to 4 years [16,34,40]. In this way, the current study is unprecedented in Brazil, as a comparative study was carried out with a control group (healthy children) aged between 5 and 11 years. Our results showed that there was no difference between healthy children and children with CHD. In contrast to findings from studies carried out in developed countries, children with heart disease did not have good performance in psychomotor development when compared with healthy children [34,36].

The demographic and socioeconomic variables identified in this study as factors associated with child neurodevelopment were consistent with those of other Brazilian studies that highlighted that environmental factors, low social class, very large families, parental criminality, and placement in foster homes are positively associated with inattention, for example [42].

Another important aspect to be mentioned is the performance of tests for the diagnosis of CHD. In the prenatal period, the unavailability of the fetal echocardiogram exam by the SUS (Brazilian public health system) was the main reason why most mothers in this study only found that their babies had heart disease after birth, a reality different from that of British fathers and mothers who learned of their babies’ illnesses while still in the womb [43]. In our study, when the fetal echocardiogram exam was recommended in pregnant women with risk factors, it was observed that, of the 13 pregnant women who underwent the exam, 9 had babies with CHD, emphasizing the importance that the exam should be made available by the health system for all mothers.

Regarding the pulse oximetry findings, which showed no difference between the two groups, it is known that the Brazilian Ministry of Health only adopted pulse oximetry in neonatal screening in 2014, and the pulse oximetry test used had a sensitivity of 75% and specificity of 99%. Therefore, it is possible that up to 25% of children with positive tests when performing echocardiography have normal results [44]. 

Although the present study provides unprecedented data on neurodevelopmental outcomes in children with cyanotic CHD, there are some limitations. The sample contained patients who were exclusively attended to by the unified health system (public) and all children studying in public schools; part of this may explain the homogeneity of the performance results from the neurodevelopmental tests. In this sense, the findings in children with better socioeconomic status may be different from those found in the present study. In addition, the attention assessment scale (SNAP IV) should also be applied to teachers; however, due to the COVID-19 pandemic and the classes being in a remote format, this was not possible. However, the children living with their parents allowed us to have a good degree of certainty in the results. Even so, this is an unprecedented study in the Brazilian population and has extreme relevance to the population in this age group (school age).

## 5. Conclusions

There was no difference in the performance of child residents of Rio Grande, Brazil, with and without cyanotic CHD in the instruments used for neurodevelopmental assessment. Among the factors associated with worse neuropsychomotor performance were socioeconomic class, age, and number of siblings. These findings demonstrate the need to examine socioeconomic determinants to improve children’s health conditions.

## Figures and Tables

**Figure 1 medicina-58-01669-f001:**
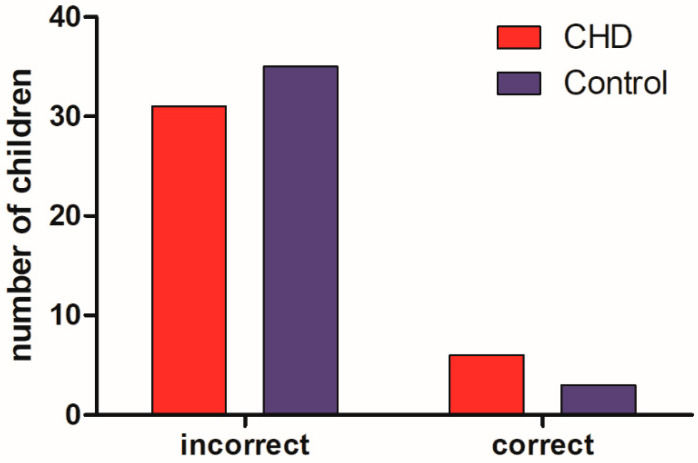
Children’s performance on the motor development scale.

**Figure 2 medicina-58-01669-f002:**
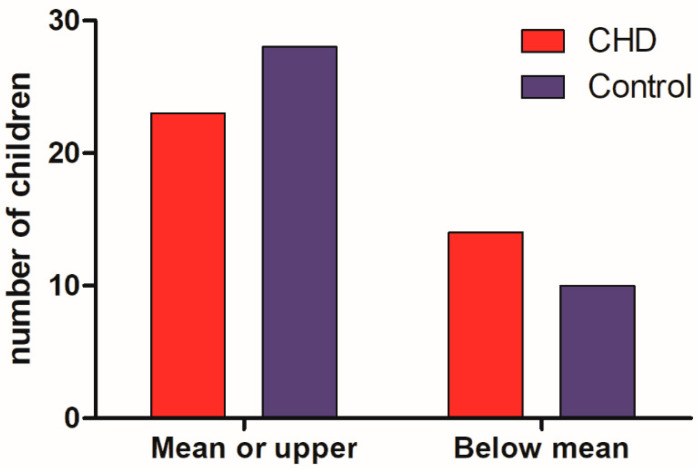
Children’s performance in the R-2 Nonverbal Intelligence test.

**Figure 3 medicina-58-01669-f003:**
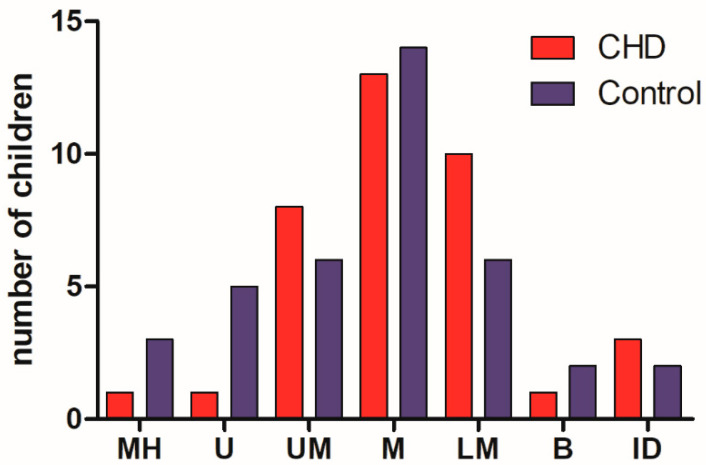
Distribution of children in intelligence classes using the R2 nonverbal intelligence test. MH: much higher; U: upper; UM: upper mean; M: mean; LM: lower mean; B: borderline; and ID: intellectually disabled.

**Figure 4 medicina-58-01669-f004:**
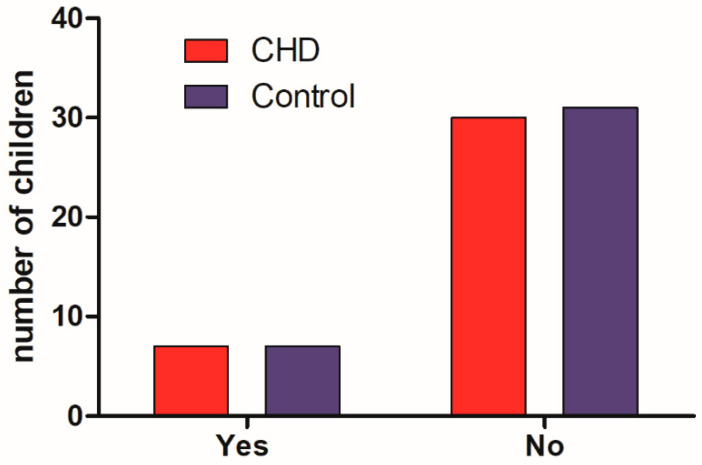
Children’s performance on the attention scale. “Yes” means that the child was categorized as having attention deficit and “No” refers to children without attention deficit.

**Table 1 medicina-58-01669-t001:** Descriptive information of the study variables.

	Total *n* (%)	CHD *n* (%)	Control *n* (%)	*p* Value
**Age (months)** (mean ± DP)	97.5 (±25.7)	95.7 (±26.5)	99.3 (±25.3)	0.54
**Socioeconomic class**				**0.01**
A-B1-B2 (upper)	11 (15)	7 (19)	4 (10)	
C1-C2	59 (79)	26 (70)	33 (87)	
D-E (lower)	5 (6)	4(11)	1 (3)	
**Gender**				0.39
Boys	44 (59)	23 (62)	21 (55)	
Girls	31 (41)	14 (38)	17 (45)	
**Number of siblings**				0.49
0	14 (19)	7 (19)	7 (18)	
1	30 (40)	14 (38)	16 (42)	
2	15 (20)	9 (24)	6 (16)	
3 or more	16 (21)	7 (19)	9 (24)	
**Reason for medical referral**				**<0.0001**
Established CHD	22 (29)	22 (59)	0 (0)	
Suspected CHD	9 (12)	7 (19)	2 (5)	
Another motive	44 (59)	8 (22)	36 (95)	
**Mother had prenatal care**				0.06
No	2 (3)	0 (0)	2 (5)	
Yes	73 (97)	37 (100)	36 (95)	
**Normal pulse oximetry**				1.00
No	7 (14)	4 (14)	3 (14)	
Yes	42 (86)	24 (86)	18 (86)	
**Fetal echocardiogram**				**0.006**
No	45 (78)	20 (69)	25 (86)	
Yes	13 (22)	9 (31)	4 (14)	

**Table 2 medicina-58-01669-t002:** Percentage of children with satisfactory performance in each of the motor tests.

	Total	CHD	Control	*p* Value
Thumb Tip	75.7	72.2	78.9	0.40
Jumps at a height of 40 cm	77.0	80.6	73.7	0.31
Squatting balance	68.9	72.2	65.8	0.44
Speed test	23.0	22.2	23.7	0.87
Recognizes human figures	59.4	55.6	63.1	0.39

**Table 3 medicina-58-01669-t003:** Crude and adjusted analysis of risk factors related to motor performance.

Variables	Crude Analysis (95% CI)	*p* Value	Adjusted Analysis (95% CI)	*p* Value
**Age**	1.001 (0.998–1.004)	0.661	1.001 (0.998–1.004)	0.571
**Socioeconomic class**	0.995 (0.978–1.013)	0.579	0.995 (0.978–1.013)	0.579
**Gender**		0.579		0.603
Girls	1.049 (0.886–1.242)		1.046 (0.883–1.240)	
Boys	1		1	
**Number of siblings**	0.996 (0.937–1.058)	0.889	0.989 (0.930–1.052)	0.730
**Reason for medical referral**		0.183		0.183
Established CHD	0.798 (0.623–1.022)		0.798 (0.623–1.022)	
Suspected CHD	0.931 (0.733–1.184)		0.931 (0.733–1.184)	
Another motive	1		1	
**Normal pulse oximetry**		0.887		0.517
Yes	1.012 (0.859–1.193)		0.953 (0.822–1.103)	
No	1		1	
**Previous admissions**		0.748		0.480
Yes	1.027 (0.873–1.207)		1.057 (0.906–1.233)	
No	1		1	

**Table 4 medicina-58-01669-t004:** Crude and adjusted analysis of risk factors related to nonverbal intelligence deficits.

Variables	Crude Analysis (95% CI)	*p* Value	Adjusted Analysis (95% CI)	*p* Value
**Age**	**0.996** **(0.993–0.999)**	**0.008 ***	**0.996** **(0.994–0.999)**	**0.012 ***
**Socioeconomic class**	**0.982** **(0.969–0.994)**	**0.005 ***	**0.983**	**0.007 ***
**Gender**		0.326		0.318
Girls	1.084 (0.923–1.273)		1.075 (0.933–1.239)	
Boys	1		1	
		0.534		0.450
**Number of siblings**	1.022 (0.960–1.096)		1.024 (0.963–1088)	
**Reason for medical referral**		0.258		0.440
Established CHD	1.143 (0.958–1.263)		1.113 (0.932–1.329)	
Suspected CHD	0.960 (0.752–1.227)		0.978 (0.802–1.193)	
Another motive	1		1	
**Normal pulse oximetry**		0.887		0.517
Yes	1.012 (0.859–1.193)		0.953 (0.822–1.103)	
No	1		1	
**Previous admissions**		0.661		0.816
Yes	1.037 (0.883–1.217)		0.983 (0.852–1.134)	
No	1		1	

* *p*-significant value.

**Table 5 medicina-58-01669-t005:** Crude and adjusted analyses of risk factors related to attention deficit.

Variables	Crude Analysis (95% CI)	*p* Value	Adjusted Analysis (95% CI)	*p* Value
**Age**	0.998 (0.978–1.018)	0.842	0.998 (0.982–1.013)	0.759
**Socioeconomic class**	**0.863** **(0.782–0.954)**	**0.004 ***	**0.868** **(0782–0.963)**	**0.008 ***
**Gender**		0.939		0.668
Girls	1.268 (0.470–3.420)		1.231 (0.477–3.172)	
Boys	1		1	
**Number of siblings**	**1.374** **(1.073–1.760)**	**0.012 ***	**1.289** **(1.043–1.593)**	**0.019 ***
**Reason for medical referral**		0.437		0.382
Established CHD	1.143 (0.374–3.491)		0.915 (0.305–2.741)	
Suspected CHD	2.095 (0.666–6.596)		2.067 (0.616–6.938)	
Another motive	1		1	
**Normal pulse oximetry**		0.123		0.066
Yes	0.473 (0.183–1.224)		0.433 (0.177–1.058)	
No	1		1	
**Previous admissions**		0.180		0.897
Yes	2.073 (0.713–6.024)		1.081 (0.330–3.541)	
No	1		1	

* *p*-significant value.

## Data Availability

Not applicable.

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
