# Peer review of "Neurodevelopmental Outcomes among Brazilian Children with Cyanotic Congenital Heart Disease and Its Associated Factors"

_medicina, 2022, doi:10.3390/medicina58111669_

Round 1

Reviewer 1 Report (Previous Reviewer 2)

The paper was improved, but the authors addressed only some of the suggestions requested. In particular results are unclear and confused (it would be useful to divide the results section into several parts to make this section more readable) and the discussion section doesn not adequately explain and support the findings of the study.

Morover I have some doubts about the choice of tools used to evaluate the three neurodevelopmental functions (motor development, non verbal intelligence and attention). Some of these tools are not internationally validated. Can you explain the choice of these assessment tools?

Author Response

Dear Reviewer,

We would like to thank you for your willingness to contribute to our work. The observations and suggestions pointed out were certainly very relevant to improve the final quality of the document. For each of the highlighted items, we present below our considerations and adjustments made in the new version of the article, which we hope have been sufficient to clarify your questions.

Point 1: The paper was improved, but the authors addressed only some of the suggestions requested. In particular results are unclear and confused (it would be useful to divide the results section into several parts to make this section more readable) and the discussion section does not adequately explain and support the findings of the study.

Response 1:

The results weren't as clear as they should have been, so we added an introductory paragraph and also separated the tools used into numbered sub-items. To better organize the text, we moved the tables of associated factors to the section corresponding to the outcome.

“The results were organized considering the 4 instruments used in data collection: sociodemographic questionnaire, motor development scale, R-2 nonverbal intelligence test and attention deficit test, detailed below.” (Extracted from the revised manuscript, Lines 156-158);3.1. Sociodemographic questionnaire (Extracted from the revised manuscript, Lines 159); 3.2. Motor development scale (Extracted from the revised manuscript, Lines 178); 3.3. Nonverbal intellence test (Extracted from the revised manuscript, Lines 195); 3.4. Attention deficit test (Extracted from the revised manuscript, Lines 211).

Discussion: Our study had findings contrary to those found in other parts of the world, regarding the worse performance in childhood neurodevelopmental outcomes in children with cyanotic congenital heart disease [14, 22. However, socioeconomic status appears to be an important factor for the absence of differences in the outcomes evaluated [16, 23, 24, 25]. The study population is composed only of children assisted by the Brazilian public health system and all of them study in public schools. We believe that the main findings of the article are discussed in the text. The discussion goes on to explain more about social vulnerabilities [37, 38, 39].

We believe that the suggestion of reorganizing the results will give them greater clarity and that in this way the discussion will remain faithful in relating the scientific literature to the findings. Even so, we added a sentence to introduce this problematic.

Lines 269-287. These results obtained from studies conducted in developed countries with better socioeconomic conditions were not reproduced in the present study. A reasonable line of reasoning to explain the findings of the present study is the social vulnerability of our study population that can impair or delay the development of psychomotor skills.

Point 2: Morover I have some doubts about the choice of tools used to evaluate the three neurodevelopmental functions (motor development, non verbal intelligence and attention). Some of these tools are not internationally validated. Can you explain the choice of these assessment tools?

Response 2:

The neurodevelopment tools used in the study (attention test, motor development test and intelligence test) were tools available for immediate use and easy to train and apply in a hospital environment. The research was carried out in the hospital and this environment requires that effective and easy-to-apply tools be used. In addition, we added to the text a number of other studies that applied the same tools.

2.4.1. Motor Development: “This scale is used in Brazilian and international studies [19,20,21]” (Extracted from the revised manuscript, Lines 116-117).

2.4.2. Nonverbal intelligence:“The model is validated by the Brazilian psychology council [23] and is used both in Brazilian studies [24,25,26] and in international studies [27,28].” (Extracted from the revised manuscript, Lines 127-128)

2.4.3. Attention Deficit: “The SNAP IV scale is widely used [31,32,33]for screening attention deficit hyperactivity disorder in children and was applied to parents and/or guardians in an appropriate room.” (Extracted from the revised manuscript, Lines 139-141)

Reviewer 2 Report (New Reviewer)

The paper “Neurodevelopmental outcomes among Brazilian children with 2 cyanotic congenital heart disease and its associated factors”

Is interesting, analysing neurodevelopmental outcomes in 37 children with cyanotic congenital heart disease (CHD) in comparison to 38 healthy children (age in both groups  5 and 11 years, 11 months ). They compared motor development, nonverbal intelligence and attention deficit according to the current tests and perfomed a statistical analysis of the variables.

As a result they did not found significant differences between children with CHD and the control group,, specifying that the factors for a worse performance on the intelligence test and  inattention was a low socioeconomic   class, age and a greater number of siblings.

I have only some minor queries:

Page 2  lines 91- I think it would be interesting to specify the types of cyanotic CHD and whether these children were operated or not.

Page 4, line 156 – their sociodemographic data – it should be “his”…

Line 165-  13 women underwent a fetal echocardiography – was a diagnose  been made?- please explain better.  Equally, it should be better to include the data of a fetal diagnosis  in the table 1 in page 5.

 In the table 2 in page 6 I cannot understand the meaning of “Recognition of the other” Can You explain it ?

Page 8, Table 4 – bottom- Number of brothers – should be better number of siblings, and also in Table 5.

Page 11- line 301- the point of 13 women who had fetal exam… it would be better to include even here the diagnostic conclusions of these cases.

In conclusion: I think the paper is interesting and requires in my opinion minor corrections. Then it could be  published.

Author Response

Dear Reviewer,

We would like to thank you for your willingness to contribute to our work. The observations and suggestions pointed out were certainly very relevant to improve the final quality of the document. For each of the highlighted items, we present below our considerations and adjustments made in the new version of the article, which we hope have been sufficient to clarify your questions.

 Page 2  lines 91- I think it would be interesting to specify the types of cyanotic CHD and whether these children were operated or not.

“Patients with CHD were only included in the survey after confirmation of the diagnosis by pediatric Doppler echocardiography (Tetralogy of Fallot - 37.8%; Transposition of great vessels - 18.9%; Pulmonary atresia - 16.2%; tricuspid atresia - 13.5%; tricuspid atresia with RV hypoplasia - 8.1% and cyanotic complex heart disease - 5.4%).” (Extracted from the revised manuscript, Lines 93-95)

Page 4, line 156 – their sociodemographic data – it should be “his”…

...but his sociodemographic data were included (Extracted from the revised manuscript, Line 166)

Line 165-  13 women underwent a fetal echocardiography – was a diagnose  been made?- please explain better.  Equally, it should be better to include the data of a fetal diagnosis  in the table 1 in page 5.

Of the 75 women who participated in the research, only 13 underwent fetal echocardiography during the period of pregnancy, all of them used the Brazilian Unified Health System (SUS) where fetal echocardiography exams are released only for those who have some risk factor in pregnancy such as a history family member with CC, pregnant woman with LUPUs or diabetes. The examination is not always carried out in a timely manner and the slowness of the system hinders the definition of the diagnosis.

We do not have the diagnosis of the 13 women who took the exam and participated in this study, but we do not believe that this data would have any influence on our results.

 In the table 2 in page 6 I cannot understand the meaning of “Recognition of the other” Can You explain it ?

In the motor test, one of the tasks was to recognize drawings of human figures and replicate what they were seeing. For example, after children visualized a human drawing with their left hand on their right ear, they reproduced what they were seeing.

Exchanged  for” Recognizes human figures” Table 2

Page 8, Table 4 – bottom- Number of brothers – should be better number of siblings, and also in Table 5.

Exchanged for Number of siblings in Table 3, 4 and 5

Page 11- line 301- the point of 13 women who had fetal exam… it would be better to include even here the diagnostic conclusions of these case

We do not have the diagnosis of the 13 women who took the exam and participated in this study, but we do not believe that this data would have any influence on our results

We are at your disposal to answer any new questions that may arise after the new review.

Best Regards,

Flavio Manoel Rodrigues da Silva Júnior

Round 2

Reviewer 1 Report (Previous Reviewer 2)

Although the authors have greatly improved the manuscript, I still have some doubts regarding the discussion of the results and the conclusions.

According to the authors, the lack of differences between the two groups in motor, cognitive and attention skills depends on the fact that the children of the study live in rural areas, with low socio-economic levels and in situations of social vulnerability. I think that this reflection is correct, but if this is true, the approach of the article should be changed.

For example, according to me the sentence “There was no difference in the performance of children with and without cyanotic CHD on the instruments used for neurodevelopmental assessment” in Conclusion section is not appropriate.

Author Response

Dear reviewer,

Thanks a lot for the comments. We would like to clarify an important issue in our study. Based on studies reported in the literature, our initial hypothesis was that there would be differences between children with and without congenital heart diseases in the neurodevelopmental outcomes used in the article. Interestingly, our initial hypothesis was refuted and the homogeneity of responses obtained and the low performance in the tests by the children led us to suppose that the socioeconomic and demographic vulnerability of our entire sample may be responsible for the absence of differences between the two groups analyzed (with and without CHD).
To make this clearer, we included our study hypothesis in the introduction (lines 97-100) and reinforced the condition of socioeconomic similarity of the two studied groups in the discussion (lines 278-279, 368-369). Finally, we make it clear in the conclusion that the findings refer to our study population and do not necessarily apply to children with other socioeconomic status or from other parts of the world (lines 377-378).

We believe that these changes make our objective more aligned.

This manuscript is a resubmission of an earlier submission. The following is a list of the peer review reports and author responses from that submission.

Round 1

Reviewer 1 Report

The topic addressed in this paper is very important and relevant.  Unfortunately there were flaws in the design of the project.  First and foremost, the conclusions of the study contradict the leading research in the field (ie Children with complex cyanotic congenital heart disease have increased Neurodevelopmental concerns and delays over time that need to be addressed).  This study did not show that.  Reasons for this I would postulate were not necessarily the research design (which I liked) but the patient population selected. The n was small for both cohorts.  AND, the cyanotic congenital heart disease cohort did not represent a true description of this group of patients (therefore the conclusions drawn have little significance or relevance to me).  Review of Table 1 includes "Previous admissions".  The two groups (CHD and Control) were equal.  This doesn't make sense in that all cyanotic congenital heart lesions to be included in a study like this I would have assumed have undergone a surgical or interventional procedure by the 96 months mean age of the group (with very few exceptions).  This tells me that the CHD group were no cyanotic congenital heart lesions.   Also, the control population had the same number of admissions (suggesting that these were NOT healthy controls).  "Reasons for medical referral" were also confusing with 12% with suspected cyanotic CHD at time of enrollment (and the "another motive" was unclear but included almost 1/4 of the CHD patients).  The "Normal pulse oximetry" being high in both groups would be an expected finding at this age (after 1+ surgeries).  The surprising finding was that both groups had abnormal saturations in 14% - again pointing out that the control population was not healthy. 

I do not believe conclusions can be drawn as stated comparing cyanotic congenital heart disease to controls when Table 1 has these discrepancies.

I was also at a loss with the title, I feel that there was additional words to be added (but not present).

Again, this is a great topic and needs to be studied more - but with a more clear differentiation in the study groups.

Reviewer 2 Report

The manuscript is very interesting and deserves to be revised for a possible publication. Despite this, the structure of the manuscript is very lacking. It would be useful to better re-write the methods (i.e., the exclusion and inclusion criteria for the sample are not inserted and more information is needed about the outcome measures, etc...) and also the results (it would be useful to divide the results section into several parts to make this section more readable).

Morover I have some doubts about the choice of tools used to evaluate the three neurodevelopmental functions (motor development, non verbal intelligence and attention). Some of these tools are not internationally validated. Can you explain the choice of these assessment tools?